# Diet Composition, Glucose Homeostasis, and Weight Regain in the YoYo Study

**DOI:** 10.3390/nu13072257

**Published:** 2021-06-30

**Authors:** Marleen A. van Baak, Nadia J. T. Roumans, Edwin C. M. Mariman

**Affiliations:** Department of Human Biology, NUTRIM School of Nutrition and Translational Research in Metabolism, Faculty of Health, Medicine and Life Sciences, Maastricht University Medical Centre+, 6200 MD Maastricht, The Netherlands; n.roumans@maastrichtuniversity.nl (N.J.T.R.); e.mariman@maastrichtuniversity.nl (E.C.M.M.)

**Keywords:** carbohydrate intake, fibre intake, fasting plasma glucose concentration, weight maintenance

## Abstract

Based on several randomized clinical trials, it has been suggested that baseline glucose homeostasis interacts with the influence of diet composition on weight loss and weight loss maintenance. In this secondary analysis of the YoYo study, a study investigating predictors of weight loss maintenance, we tested the hypothesis that (self-selected) dietary carbohydrate and/or fibre intake interact with the glucose homeostasis parameters for weight loss maintenance. Sixty-one overweight or obese individuals lost around 10 kg of body weight on an energy-restricted diet and were then followed for 9 months. During this period, participants were advised to maintain their body weight and eat a healthy diet without further recommendations on calorie intake or diet composition. Contrary to our hypothesis, carbohydrate intake showed no positive association with weight regain after weight loss, and no interaction with baseline fasting glucose concentration was found. There was a non-significant negative association between fibre intake and weight regain (B = −0.274, standard error (SE) 0.158, *p* = 0.090), but again, no interaction with fasting plasma glucose was found. In conclusion, the data from the YoYo study do not support a role for baseline glucose homeostasis in determining the association between self-reported carbohydrate and/or fibre intake and weight regain after weight loss.

## 1. Introduction

Because of the high prevalence of weight regain after successful weight loss [1,2,3,4], prevention of weight regain is a major challenge in the management of obesity. The biological mechanisms underlying weight regain are currently being studied in order to develop strategies to improve long-term weight loss success [5,6,7]. Many different factors in different combinations are likely to play a role. This suggests that personalized approaches may be necessary to more effectively tackle the problem of weight regain after weight loss.

In this context, the potential role of diet composition has been investigated. Hjorth and colleagues recently published a series of articles [8,9,10,11,12] in which they provided evidence that the response of weight loss and weight loss maintenance on diets differing in glycaemic load depends on glucose homeostasis: individuals with elevated fasting plasma glucose (FPG) values (prediabetic individuals) respond better (more weight loss and less weight regain) to diets high in fibre and with a low glycaemic index (GI), whereas individuals with high FPG values (mostly individuals with type 2 diabetes mellitus (T2DM)) should also reduce their total carbohydrate intake and increase their fat intake [12].

The YoYo study is a randomized controlled trial (RCT) which examined the effect of different rates of weight loss on weight loss maintenance and the molecular changes in adipose tissue that might predict long-term weight maintenance [4,13,14,15,16]. In this secondary analysis of the YoYo study, we investigated whether the self-selected carbohydrate content of the diet during the weight maintenance period after a period of weight loss influenced weight regain depending on the basal glucose homeostasis of the participants. We hypothesized that there is a positive association between dietary carbohydrate content and weight regain after weight loss and/or a negative association between fibre intake and weight regain after weight loss, which is modified by glucose homeostasis parameters.

## 2. Materials and Methods

The design and methods of the YoYo study have previously been described in detail [4,16]. The trial was registered with www.clinicaltrials.gov (accessed on 29 June 2021) as NCT01559415.

### 2.1. Participants

In summary, 61 individuals that were overweight or obese (body mass index (BMI) 28–35 kg/m^2^) were included in the study. They were healthy and all had fasting plasma glucose concentrations ≤6.1 mmol/L at screening. All participants gave informed consent before entering the study. The study was performed according to the Declaration of Helsinki and was approved by the Medical Ethics Committee of Maastricht University Medical Centre+.

### 2.2. Study Design

After baseline measurements, participants were randomized into two groups: slow weight loss and rapid weight loss. In the slow weight loss group, participants underwent a 12-week low calorie diet (LCD) intervention (1250 kcal/day for 12 weeks). Those in the rapid weight loss group were provided with a very-low-calorie liquid diet (VLCD) (500 kcal/day) combined with an unrestricted amount of low-calorie vegetables for 5 weeks. Subsequently, both groups were prescribed a weight maintenance diet for 4 weeks based on their individual energy requirements with a composition according to Dutch healthy diet recommendations. Thereafter, the participants were followed for another 9 months. No dietary advice was given during this period to mimic non-restricted free-living conditions. Baseline measurements were repeated at the end of the weight loss (WL) phase, at the end of the subsequent 4-week controlled weight maintenance (CWM) phase, and after the 9 months of follow-up (FU).

### 2.3. Measurements

At each of the time points indicated above, the following measurements were performed: weight, height, body composition, and a fasting venous blood sample were collected for measurement of plasma glucose and insulin concentrations.

The participants were weighed on the same scale (Seca model 861, Hamburg, Germany) accurate to the nearest 0.1 kg in light clothing after an overnight fast of at least 10 h. Body volume was determined with air-displacement plethysmography using the Bod Pod device (Cosmed, Rome, Italy) according to the manufacturer’s instructions and as described by Dempster and Aitkens [17]. The thoracic gas volume was predicted using the equations incorporated in the Bod Pod software. Body density, as calculated by the Bod Pod, was used to calculate body composition according to the two-compartment model as described by Siri [18].

Plasma glucose was measured by a standard enzymatic assay and plasma insulin by means of a commercially available radio-immunoassay kit (Human insulin specific RIA Millipore Corporation, Billerica, MA, USA). HOMA-IR (homeostatic model assessment for insulin resistance) was calculated from fasting glucose and insulin values (HOMA-IR = (glucose (mM) × insulin (mM))/22.5).

### 2.4. Dietary Intake

Three-day weighed dietary records (including one non-working day) were obtained at three time points (baseline, end of controlled weight maintenance phase (CWM), and at follow-up (FU)). Before calculating the nutrient intake, diaries were checked by the dietician, and implausible values were discussed with the participant and corrected where appropriate.

All foods reported were linked to the 2011 Dutch food consumption table [19], and nutrient intakes were calculated and averaged over the three recorded days. The nutrient intake values obtained at the end of the CWM period and at FU were averaged to reflect dietary intake over the follow-up period. Since we do not know the time course of potential dietary changes over time, we considered this the best approach. If dietary data of one of the two time points were missing, we used the data from the available time point. We also analysed the data for the two time points separately.

### 2.5. Data Analysis and Statistics

Data are presented as mean ± standard deviation (SD), unless indicated otherwise. In the analyses, no distinction was made between the two groups (VLCD and LCD). Repeated measurements analysis of variance (ANOVA) was used to compare different time points. Linear regression analysis was used to analyse the association between weight regain as the dependent variable and glucose homeostasis parameters and diet composition as the independent variables. Mixed-models ANOVA was used to analyse differences in weight regain between specific groups of participants. SPSS version 25 was used for the analyses.

## 3. Results

Four participants withdrew from the study—two because of cancer (both VLCD group) and two because of personal circumstances (one in each group). Therefore, data of the remaining 57 participants were evaluated. Some blood samples and food diaries were missing at some time points and, therefore, the sample size slightly varies according to the type of analysis performed. Data on anthropometrics, glucose homeostasis parameters, and macronutrient intakes are shown in Table 1.

Participants lost around 9 kg during the period of energy restriction, remained weight stable during the controlled weight maintenance period, and regained half of their weight loss during the 9-month follow-up period. Percentage body fat showed a similar pattern, but during the controlled weight maintenance period there was a small but significant further reduction in body fat despite the unchanged body weight. Fasting glucose, insulin, and HOMA-IR were reduced during weight loss and did not significantly change thereafter.

Energy intake was significantly lower over the follow-up period than at baseline. Fat intake, expressed as percentage of energy intake, was lower during the follow-up period than at baseline, whereas protein intake and fibre intake were higher. Carbohydrate intake did not significantly change over the course of the study; if anything, it slightly increased. During the follow-up period, carbohydrate (CHO) intake was negatively correlated with both fat intake (*r* = −0.654, *p* = 0.000) and protein intake (*r* = −0.446, *p* = 0.001), but not with fibre intake (*r* = 0.045, *p* = 0.748). Fibre intake was positively correlated with protein intake (*r* = 0.419, *p* = 0.002), but not with fat intake (*r* = −0.259, *p* = 0.059).

In the total group of participants, there was no association between baseline fasting glucose concentration and weight regain (Figure 1, Table 2). Similar results were obtained for baseline fasting insulin and HOMA-IR and for the glucose homeostasis parameters at the end of the CWM period. Adjusting for sex did not change the outcomes (data not shown).

Next, we analysed the association between the dietary CHO intake during the follow-up period and weight regain (Figure 2). The relationship was not statistically significant (*p* = 0.896). When the regression coefficients of the relationship between carbohydrate intake and weight regain were compared in the participants with baseline FPG ≤ 5.6 mmol/L and those >5.6 mmol/L, the regression coefficients were not statistically significant (B = 0.113, SE 0.087, *p* = 0.205 vs. B = −0.270, SE 0.155, *p* = 0.121) (Figure 3).

When participants were divided into groups based on their baseline fasting glucose concentration (< or ≥5.6 mmol/L) and their carbohydrate intake (≤ or >the median), there were 20 participants in group 1 (FPG <5.6 and ≤ median CHO intake (48.7 energy%), 19 in group 2 (FPG < 5.6 mmol/L and >median CHO intake, 6 in group 3 (FPG ≥ 5.6 mmol/L and ≤ median CHO intake), and 4 in group 4 (FPG ≥ 5.6 mmol/L and >median CHO intake). Weight regain was 4.5 ± 3.6 kg in group 1, 4.5 ± 3.3 kg in group 2, 5.8 ± 4.2 kg in group 3, and 3.0 ± 3.0 kg in group 4. Weight regain was not significantly different among the groups (ANOVA, *p* = 0.697, *n* = 47) (Figure 4). Removing the outlier in group 2 did not change this result.

Similar analyses were performed for fibre intake. Fibre intake during FU tended to be negatively associated with weight regain (B = −0.274, SE 0.158, *p* = 0.090) (Figure 5). However, baseline FPG concentration did not modify this relationship significantly (Figure 6). In addition, when participants were grouped according to baseline FPG (≤5.6 mmol/L vs. > 5.6 mmol/L) and fibre intake during FU (below or above the median (13.5 g/1000 kcal), no differences were found among the groups (ANOVA, *p* = 0.287) (Figure 7). There were 18 subjects in the group with FPG ≤ 5.6 mmol/L and below median fibre intake, 21 in the group with FPG ≤ 5.6 mmol/L and above median fibre intake, 7 in the group with FPG > 5.6 mmol/L and below median fibre intake, and 3 in the group with FPG > 5.6 mmol/L and above median fibre intake. After removal of the two outliers, the between-group difference remained statistically insignificant (*p* = 0.388).

We repeated all analyses for carbohydrate and fibre intake at the separate time points (end of CWM and end of FU). The results were comparable with the results for the mean of intakes at the end of CWM and end of FU reported above. We also performed analyses for the association between intake of mono and disaccharides and weight regain and its interaction with FPG. Results were similar to those for total CHO intake. In addition, we analysed the CHO/fibre intake ratio which also did not result in significant outcomes (see Appendix A).

## 4. Discussion

Contrary to our hypothesis, we found no positive association between weight regain and baseline or post-weight loss parameters of glucose homeostasis (FPG, fasting insulin (FI), and HOMA-IR) and no modification by carbohydrate intake or fibre intake during the period of weight regain. In contrast, the participants with FPG > 5.6 mmol/L showed a negative association between carbohydrate intake and weight regain. Although there was a non-significant negative association between fibre intake and weight regain, no interaction with FPG was found. Thus, we were unable to confirm the previous analyses by Hjorth et al. which suggested that individuals with elevated fasting plasma glucose (FPG) values (prediabetic individuals) respond better to diets high in fibre and with a low glycaemic index (GI) [12].

However, there are several differences with the studies on which Hjorth et al. based their conclusion [8,9,10,11] that need to be addressed. With respect to the design of the studies, the YoYo study most resembles the DiOGenes study [11,20] and the Monounsaturated Fatty Acids in Obesity study [10,21] in that an ad libitum diet intervention followed an initial weight loss phase. The other studies were either combinations with energy-restriction or were not weight loss studies [9,11,12]. One obvious difference between the YoYo study and the DiOGenes and Monounsaturated Fatty Acids in Obesity studies is that, although our study was an RCT, we did not randomise the participants with respect to diet composition. As a result, it should be regarded as an observational study with all the limitations associated with observational studies. Furthermore, the (type of) carbohydrate and fibre intake were self-chosen by the participants. They only received information about recommendations for a healthy Dutch diet. With respect to carbohydrates and fibre, this advice recommends ample daily consumption of fruits, vegetables, and whole-grain products and to limit the intake of foods and drinks with sugars [22]. Therefore, the range of CHO intake (35–62 energy%) and fibre intake (6–20 g/1000 kcal) is what can be expected in the context of the habitual Dutch diet and is not clearly different from the intakes in the DiOGenes study. We were not able to make the distinction according to glycaemic load by also including the glycaemic index of the CHOs consumed in our analysis, as in the DiOGenes trial, because these data were not available for the YoYo study. However, it seems unlikely that those with a high CHO intake consumed predominantly low GI foods, whereas those with a low CHO intake consumed mainly high GI products. In the Monounsaturated Fatty Acids in Obesity study, FPG affected the difference in weight regain response between the low fat/high fibre diet and the average Danish diet. The low fat/high fibre diet group had on average a higher total carbohydrate intake (57.6 energy%) and a higher fibre intake (16.7 g/1000 kcal) than the average Danish diet group (49.8 energy% and 12.1 g/1000 kcal, respectively) [21]. In the YoYo study, the average carbohydrate intake was comparable and the fibre intake slightly higher than in the average Danish diet group, but both were lower than in the low fat/high fibre group. Although the total range of fibre intakes in the YoYo study covered the ranges reported in the Monounsaturated Fatty Acids in Obesity study, limited values at both ends of the distribution may explain the less strong association between fibre intake and weight regain.

Another difference is that the YoYo study included equal numbers of males and females, whereas in the DiOGenes study, only 34% of the participants were male [20]. In the Monounsaturated Fatty Acids in Obesity study, 45% of the participants were male [21]. However, including sex in our regression analyses did not change the outcome. The females in our study were likely to be partly pre- and partly post-menopausal. We do not have data on menopause status, so we were unable to look into a potential effect of hormonal status.

An obvious limitation of this study is the reliance on self-reported dietary intakes, which are known to have low reliability. Another important limitation is the sample size of the YoYo study, which was relatively small, which increases the risk of failure to detect existing differences. However, the fact that weight regain, if anything, was lower and the association with carbohydrate intake was negative in the group with higher FPG suggests that this risk was probably not very high for this analysis. For the association between fibre intake and weight regain, we cannot fully exclude this possibility, although an interaction with FPG is unlikely.

## 5. Conclusions

In conclusion, in this secondary analysis of the YoYo study, we did not find evidence for a relationship between the fasting plasma glucose concentration of the participants at baseline and weight regain after a period of weight loss nor for an interaction with the carbohydrate content or fibre content of the diet. However, given the observational nature of the study, its relatively small sample size, and its reliance on self-reported dietary data, it seems important to pursue the idea of an interaction between glucose homeostasis parameters and diet composition in body weight regulation in additional studies and further explore the exact relationships.

## Figures and Tables

**Figure 1 nutrients-13-02257-f001:**
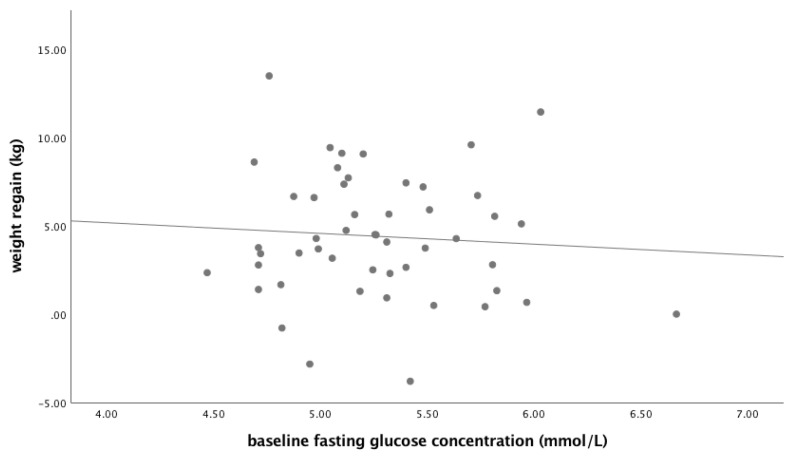
Association between baseline fasting glucose concentration and weight regain (*n* = 50).

**Figure 2 nutrients-13-02257-f002:**
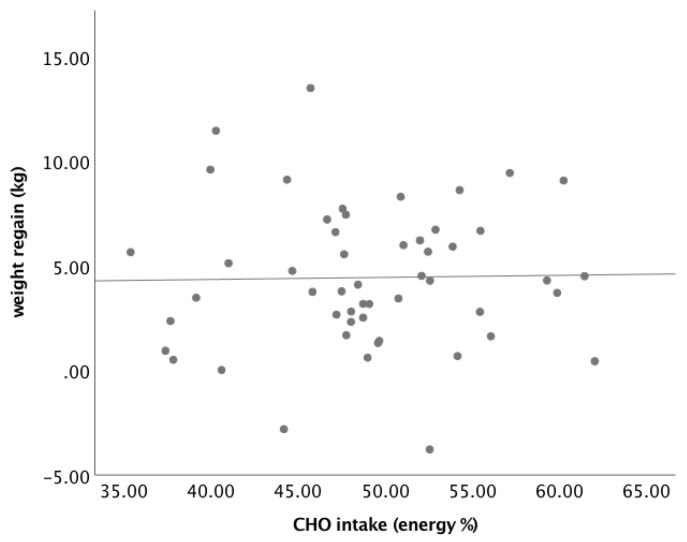
Association between dietary carbohydrate (CHO) intake during follow-up (FU) and weight regain (*n* = 52).

**Figure 3 nutrients-13-02257-f003:**
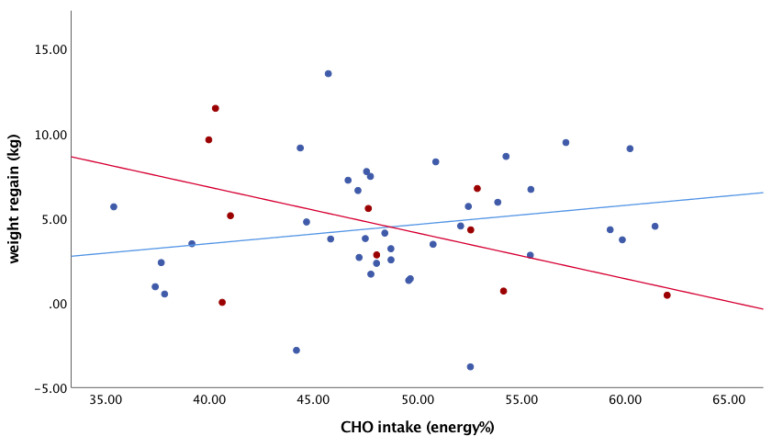
Associations between carbohydrate intake and weight regain in the participants with baseline normoglycemia (≤5.6 mmol/L) and those with baseline elevated fasting plasma glucose (FPG) (>5.6 mmol/L). Blue dots/line: FPG ≤ 5.6 mmol/L; red dots/line: FPG > 5.6 mmol/L (*n* = 49).

**Figure 4 nutrients-13-02257-f004:**
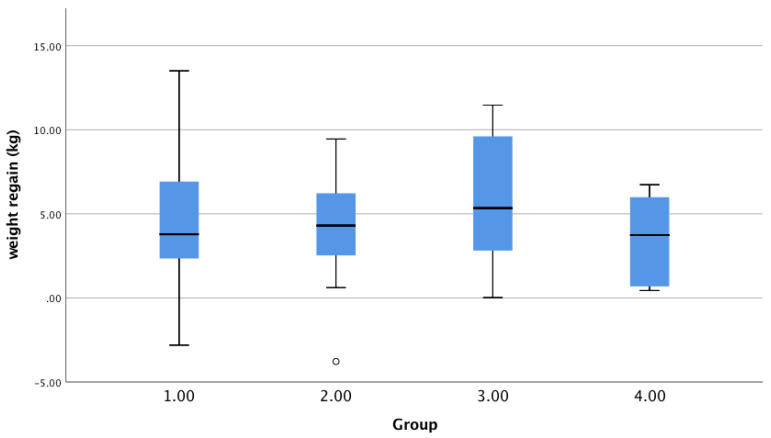
Weight regain in groups of subjects with different combinations of baseline FPG and CHO intake. Group 1: baseline FPG < 5.6 mmol/L and CHO intake during the FU period ≤ median; group 2: FPG < 5.6 mmol/L and >median CHO intake (48.7 energy%); group 3: FPG ≥ 5.6 mmol/L and ≤ median CHO intake; group 4: FPG ≥ 5.6 mmol/L and >median CHO intake (*n* = 49).

**Figure 5 nutrients-13-02257-f005:**
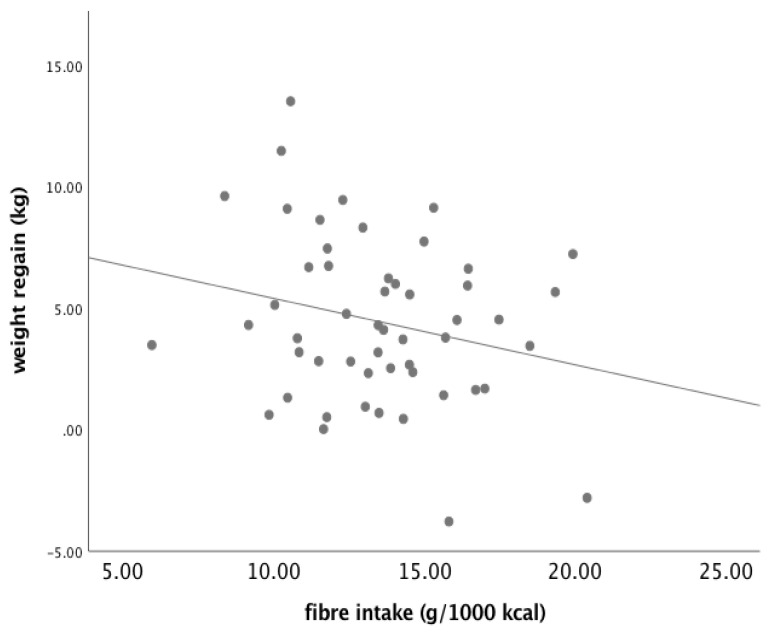
Association between dietary fibre intake and weight regain (*n* = 49).

**Figure 6 nutrients-13-02257-f006:**
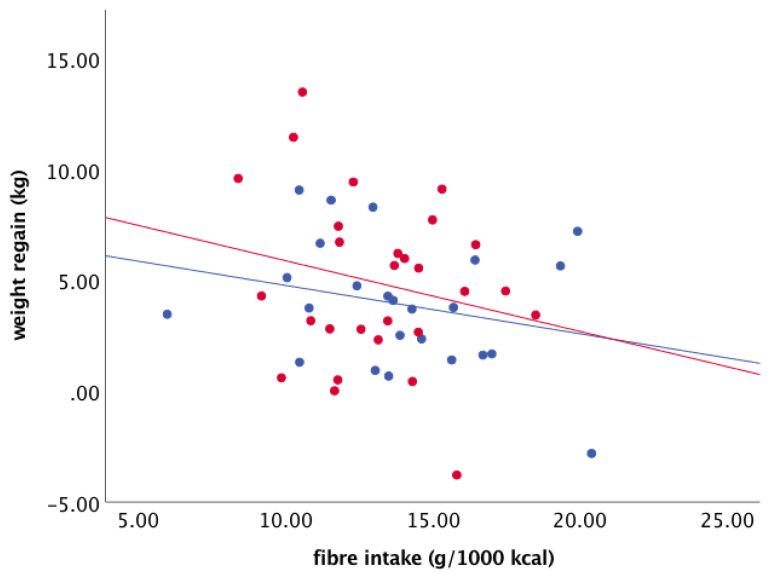
Associations between fibre intake during FU and weight regain in the participants with normoglycemia (≤5.6 mmol/L) and those with elevated FPG (>5.6 mmol/L) at baseline. Blue dots/line: FPG ≤ 5.6 mmol/L; red dots/line: FPG > 5.6 mmol/L (*n* = 49).

**Figure 7 nutrients-13-02257-f007:**
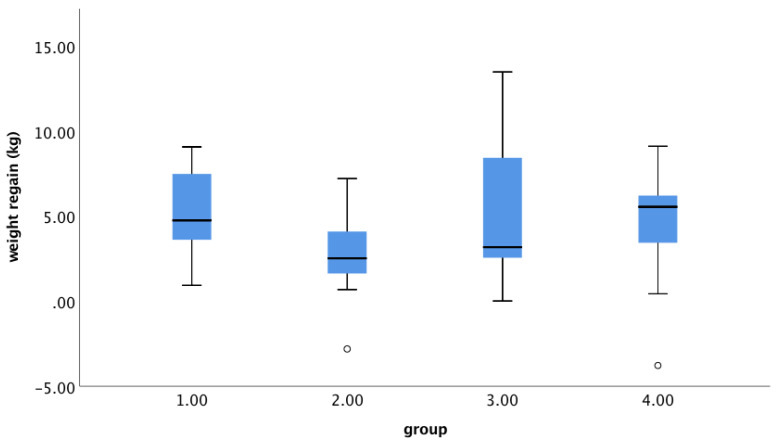
Weight regain in groups of subjects with different combinations of baseline FPG and fibre intake during the FU period. Group 1: baseline FPG < 5.6 mmol/L and fibre intake ≤ median (13.5 g/1000 kcal); group 2: FPG < 5.6 mmol/L and > median fibre intake; group 3: FPG ≥ 5.6 mmol/L and ≤ median fibre intake; group 4: FPG ≥ 5.6 mmol/L and > median fibre intake (*n* = 49).

**Table 1 nutrients-13-02257-t001:** Subject characteristics and anthropometrics, glucose homeostasis parameters, and dietary intakes (mean (SD)) at the different time points of the study (baseline, end of WL, end of CWM, and FU).

	Baseline	End of WL	End of CWM	FU	*p* Value *
Age (year)	51.3 (9.1) (*n* = 57)				
Sex (male/female)	27/30				
Height (cm)	172.3 (8.9) (*n* = 57)				
Weight (kg)	92.5 (9.9) (*n* = 57)	83.9 (9.2) ^a^ (*n* = 57)	83.7 (9.5) ^a^ (*n* = 57)	88.3 (10.2) ^a,c,d^ (*n* = 55)	<0.001 (*n* = 55)
BMI (kg/m^2^)	31.2 (2.3) (*n* = 57)	28.3 (2.4) ^a^ (*n* = 57)	28.2 (2.4) ^a^ (*n* = 57)	29.6 (2.7) ^a,c,d^ (*n* = 55)	<0.001 (*n* = 55)
Body fat%	39.8 (8.6) (*n* = 55)	34.7 (10.2) ^a^ (*n* = 55)	33.8 (10.3) ^a,c^ (*n* = 55)	36.3 (9.8) ^a,c,d^ (*n* = 53)	<0.001 (*n* = 53)
FPG (mmol/L)	5.25 (0.43) (*n* = 52)	4.96 (0.40) ^a^ (*n* = 52)	5.06 (0.44) ^a^ (*n* = 52)	4.96 (0.44) ^a^ (*n* = 52)	<0.001 (*n* = 52)
FPI (µIU/mL)	16.2 (6.0) (*n* = 52)	11.7 (4.8) ^a^ (*n* = 52)	13.3 (5.9) ^a^ (*n* = 52)	13.4 (8.6) ^b^ (*n* = 52)	<0.001 (*n* = 52)
HOMA-IR	3.84 (1.59) (*n* = 52)	2.65 (1.21) ^a^ (*n* = 52)	2.98 (1.34) ^a^ (*n* = 52)	2.85 (1.85) ^a^ (*n* = 52)	<0.001 (*n* = 52)
Energy intake (kJ/day)	8910 (3163) (*n* = 55)	NR	6904 (1944) ^a^ (*n* = 53)	7553 (2460) ^b^ (*n* = 44)	0.002 (*n* = 42)
Carbohydrate intake (energy%)	46.2 (7.8) (*n* = 55)	NR	49.1 (7.1) (*n* = 53)	49.2 (7.5) (*n* = 44)	0.092 (*n* = 42)
Fat intake (energy%)	36.1 (6.8) (*n* = 55)	NR	30.6 (6.2) ^a^ (*n* = 53)	32.1 (6.3) ^b^ (*n* = 44)	<0.001 (*n* = 42)
Protein intake (energy%)	17.6 (4.1) (*n* = 55)	NR	21.3 (4.7) ^a^ (*n* = 53)	18.8 (4.2) ^b,d^ (*n* = 44)	<0.001 (*n* = 42)
Fibre intake (g/1000 kcal)	10.4 (3.6) (*n* = 55)	NR	14.4 (4.1) ^a^ (*n* = 53)	12.6 (3.5) ^b,d^ (*n* = 44)	<0.001 (*n* = 42)

WL = weight loss; CWM = controlled weight maintenance; FU = follow-up; *n* = number of subjects; BMI = body mass index; FPG = fasting plasma glucose; FPI = fasting plasma insulin concentration; HOMA-IR = homeostatic model assessment of insulin resistance; * *p* value of repeated measurements ANOVA and N; NR= not recorded. Post-hoc pairwise comparison (Bonferroni corrected): ^a^ *p* < 0.01 vs. baseline; ^b^ *p* < 0.05 vs. baseline; ^c^ *p* < 0.01 vs. end of weight loss; ^d^ *p* < 0.01 vs. end of controlled weight maintenance.

**Table 2 nutrients-13-02257-t002:** Results of linear regression analysis with weight regain (kg) as the dependent variable and glucose homeostasis parameters as the independent variables (*n* = 50).

	B (SE)	*p* Value
Baseline FPG (mmol/L)	−0.61 (1.17)	0.604
Baseline FPI (µIU/mL)	0.09 (0.08)	0.312
Baseline HOMA-IR	0.29 (0.31)	0.361
End of CWM FPG (mmol/L)	−0.70 (1.14)	0.540
End of CWM FPI (µIU/mL)	0.12 (0.09)	0.208
End of CWM HOMA-IR	0.42 (0.37)	0.264

CWM = controlled weight maintenance; FPG = fasting plasma glucose concentration; FPI = fasting plasma insulin concentration; HOMA-IR = homeostatic model assessment of insulin resistance; B (SE) = regression coefficient (standard error).

## Data Availability

Data reported in this article are not publicly available, but may be obtained from the authors upon request.

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
