# Peer review of "Diet Composition, Glucose Homeostasis, and Weight Regain in the YoYo Study"

_nutrients, 2021, doi:10.3390/nu13072257_

Round 1
Reviewer 1 Report
Line 56, add “content” after dietary carbohydrate.
Line 168, add median carbohydrate intake in number.
Line 186, add median fiber intake in number.
Table 1, intakes of carbohydrate, fat, protein (energy %) and fiber intake in Table 1 are different from the data published in Nutrients 2017, 9, 1205; doi:10.3390/nu9111205, although they are the same study and the same authors, and used the same data obtained from the same participants.
Need more detailed introduction.
Table 1, add subcategory based on FPG; FPG <5.6 mmol/L and FPG ≥5.6 mmol/L and add their data.
The authors did a good job comparing their data with the previous studies. However, they need extensive discussion to explain their data/results.
Analyze the following data: 1) association between dietary sugar intake during FU and weight regain; 2) associations between sugar intake and weight regain based on participants’ fasting plasma glucose level; 3) weight regain in groups of subjects with different combinations of baseline FPG and sugar intake.
Author Response
We thank the reviewer for the helpful comments, which we will address post-by-point below.
Line 56, add “content” after dietary carbohydrate.
Done
Line 168, add median carbohydrate intake in number.
We have added the median carbohydrate intake (48.7 energy%) to line 168 (now line 179) and the legend of Figure 4.
Line 186, add median fiber intake in number.
We have added the median fibre intake (13.5 g/1000 kcal) to line 186 (now line 199) and the legend of Figure 7.
Table 1, intakes of carbohydrate, fat, protein (energy %) and fiber intake in Table 1 are different from the data published in Nutrients 2017, 9, 1205; doi:10.3390/nu9111205, although they are the same study and the same authors, and used the same data obtained from the same participants.
We understand the confusion. The dietary data published in Nutrients 2017, 9, 1205; doi:10.3390/nu9111205 were the mean values of the data at the end of the controlled weight maintenance phase and the follow-up measurement. In the supplemental material of that paper the values at the separate time points were presented. Here we reported the data also separately for each time point, but only for those that had complete data at all time points. These data were used in the repeated measurements ANOVA to analyse changes over time. Therefore, the numbers vary between the two tables. To avoid this confusion, we have adapted table 1 to report all available data at each time point and added the sample size on which the repeated measurements ANOVA was based.
Need more detailed introduction.
We prefer to keep the introduction short and concise, but if there is specific information that the reviewer is missing, we will consider adding it.
Table 1, add subcategory based on FPG; FPG <5.6 mmol/L and FPG ≥5.6 mmol/L and add their data.
We have added a table with the descriptive data of Table 1 for the two subgroups as supplemental material (Supplemental Table 1).
The authors did a good job comparing their data with the previous studies. However, they need extensive discussion to explain their data/results.
Analyze the following data: 1) association between dietary sugar intake during FU and weight regain; 2) associations between sugar intake and weight regain based on participants’ fasting plasma glucose level; 3) weight regain in groups of subjects with different combinations of baseline FPG and sugar intake.
We did the analyses that you suggested, but findings were similar to those for total carbohydrate content: no significant association between dietary sugar intake during FU and weight regain, no difference when fasting plasma glucose (FPG) subgroups were compared and no differences in weight regain with the different combinations of FPG and sugar intake.
We have mentioned this in the Discussion and added the results in the Supplemental material (Supplemental figures 5 to 7).
Reviewer 2 Report
This is an interesting study aimed to evaluate the hypothesis that dietary carbohydrates and fibre intake interact with glucose homeostasis and influence weight regain. The results obtained did not support such hypothesis and no influence of carbohydrate intake on weight regain was evidenced.
This study supports the need of further, controlled trials to define the real role of carbohydrates and, in general, of diet composition in weight loss e weight regain.
The study design was fine and the parameters evaluated appear to be appropriate.
One point deserves a clarification by the authors:
in the sample were included thirty women age range 42-60 more or less. This means that in the sample coexist women very different one from another in terms of hormonal and metabolic status. Could the authors provide the number of fertile and menopausal/postmenopausal women present in the sample? Could they consider and discuss this point
Author Response
We thank the reviewer for the positive comments.
The main concern of the reviewer is the possible heterogeneity of the female population given the range in age. The metabolic status has been considered to a certain extent by the metabolic measurements that were performed in fasting plasma. However, we do not have any information about the hormonal status of our female participants. We can only speculate about the potential influence of hormonal status on weight regain and its association with the dietary carbohydrate intake and the interaction with glucose homeostasis. We checked whether there were any differences in the analyses we performed between men and women, as already indicated in the Discussion, which was not the case.
We have added the following to the Discussion:
Lines 267-269: The females in our study were likely to be partly pre- and partly post-menopausal. We do not have data on menopause status, so we were unable to look into a potential effect of hormonal status.
Reviewer 3 Report
Interesting study, however, it is always difficult to know whether the self-reported diet data are accurate enough (due to the inherent limitations of the available tools) to detect any associations in such small sample size.
Abstract
Reword the "trend" to a non-significant difference. A p-value 0.09 is not a trend. (same in the discussion)
Add "self-reported" before carbohydrate intake in the last sentence
Methods
The authors need to strongly justify why they averaged the nutrient data at the two follow-up points. Seeing the average follow-up effect is not a sufficient justification. These two time points are quite different (the one is after a very prescriptive diet and the other after a free-living diet). I suggest that you repeat your analysis separately for each time point.
Substantial more detail needs to be added in the dietary intake analysis. Please expand on whether this was a 24h recall or prospective daily, detailed procedures (semi-quantitative, quantitative), how you handled data cleaning, whether the 3 daily records at each timepoint were averaged or weighted based on some factor, etc.
Worth making explicit that you had complete data for all 57 participants or whether you imputed any.
As you acknowledge, you were not able to assess glycaemic load. This is an important limitation. However, a very useful analysis I suggest to try and address this point is to assess the effect of the carbohydrate:fibre ratio. See Mozaffarian RS et al, Publlc Health Nutrition 2013 https://www.cambridge.org/core/journals/public-health-nutrition/article/identifying-whole-grain-foods-a-comparison-of-different-approaches-for-selecting-more-healthful-whole-grain-products/A890CFF17F65C81B03EEA929A23C5F2E
Table 1: p-values of 0.000 must be reformatted to <0.001.
Suggest you really tone down the statement about the possibility that the study was powered for the CHO analysis. You don't know this (unless you provide readers with post-hoc power calculations).
Self-report of diet data is another limitation to acknowledge.
Conclusion: I don't understand the conclusion. As you have not found evidence of an effect, why do you think that it seems important to continue pursuing this idea? Should we instead disregard it and focus on other ideas (e.g. total energy intake/expenditure)?
conclusion: typo: additional
Author Response
Thank you for your helpful comments, which we will address point-by-point below.
Interesting study, however, it is always difficult to know whether the self-reported diet data are accurate enough (due to the inherent limitations of the available tools) to detect any associations in such small sample size.
We agree with this comment and added this as a limitation of the study.
Abstract
Reword the "trend" to a non-significant difference. A p-value 0.09 is not a trend. (same in the discussion)
Done
Add "self-reported" before carbohydrate intake in the last sentence
Done
Methods
The authors need to strongly justify why they averaged the nutrient data at the two follow-up points. Seeing the average follow-up effect is not a sufficient justification. These two time points are quite different (the one is after a very prescriptive diet and the other after a free-living diet). I suggest that you repeat your analysis separately for each time point.
We have added the following to the Methods:
lines 105-107: Since we do not know the time course of potential dietary changes over time, we considered this the best approach. However we also analysed the data for the two time points separately.
We have repeated the analyses for the dietary intakes at each time point and added them as Supplemental material (Supplemental figures 1 to 4). This has been mentioned in the text (lines 216-218).
Substantial more detail needs to be added in the dietary intake analysis. Please expand on whether this was a 24h recall or prospective daily, detailed procedures (semi-quantitative, quantitative), how you handled data cleaning, whether the 3 daily records at each timepoint were averaged or weighted based on some factor, etc.
Dietary intake was assessed by weighed 3-d food diaries. Before calculating the nutrient intake, diaries were checked by the dietician and implausible values were discussed with the participant and corrected where appropriate. Intake data were averaged over the three days. This has been added to the Methods (lines 97-108).
Worth making explicit that you had complete data for all 57 participants or whether you imputed any.
We don't have all data complete for the 57 participants that completed the study. This has been mentioned in the Results (lines 120-122). We took care to add the number of participants in each analysis. We did not impute data, except when calculating the mean dietary intake over the follow-up period (lines 106-107).
As you acknowledge, you were not able to assess glycaemic load. This is an important limitation. However, a very useful analysis I suggest to try and address this point is to assess the effect of the carbohydrate:fibre ratio. See Mozaffarian RS et al, Publlc Health Nutrition 2013 https://www.cambridge.org/core/journals/public-health-nutrition/article/identifying-whole-grain-foods-a-comparison-of-different-approaches-for-selecting-more-healthful-whole-grain-products/A890CFF17F65C81B03EEA929A23C5F2E
We have analysed the CHO/fibre ratio as suggested, the analyses have been added to the Supplemental material (Supplemental figures 8 to 10). No significant results were found.
Table 1: p-values of 0.000 must be reformatted to <0.001.
Done
Suggest you really tone down the statement about the possibility that the study was powered for the CHO analysis. You don't know this (unless you provide readers with post-hoc power calculations).
We have deleted this statement.
Self-report of diet data is another limitation to acknowledge.
We have acknowledged this in the discussion:
Lines 254-255: An obvious limitation of this study is the reliance on self-reported dietary intakes, which are known to have low reliability.
Conclusion: I don't understand the conclusion. As you have not found evidence of an effect, why do you think that it seems important to continue pursuing this idea? Should we instead disregard it and focus on other ideas (e.g. total energy intake/expenditure)?
Because to our knowledge this is the first study not in line with the studies by Hjorth et al and given the limitations of our study (observational, small sample size, self-reported dietary intake data), we think the conclusion is justified. We have slightly modified the conclusion:
Lines 266-270: However, given the observational nature of the study, its relatively small sample size and reliance on self-reported dietary data, it seems important to pursue the idea of an interaction between glucose homeostasis parameters and diet composition in body weight regulation in additional studies and further explore the exact relationships.
conclusion: typo: additional
corrected
Round 2
Reviewer 3 Report
happy with the revisions
One typo: dietitian should be spelled with a t, not a c.